# A Joint Extraction System Based on Conditional Layer Normalization for Health Monitoring

**DOI:** 10.3390/s23104812

**Published:** 2023-05-16

**Authors:** Binbin Shi, Rongli Fan, Lijuan Zhang, Jie Huang, Neal Xiong, Athanasios Vasilakos, Jian Wan, Lei Zhang

**Affiliations:** 1School of Biological and Chemical Engineering, Zhejiang University of Science and Technology, Hangzhou 310023, China; sbb_edu_stu@126.com (B.S.); fanrongli@zust.edu.cn (R.F.); 2School of Information and Electronic Engineering, Zhejiang University of Science and Technology, Hangzhou 310023, China; 121107@zust.edu.cn (L.Z.); huangjie@zust.edu.cn (J.H.); wanjian@zust.edu.cn (J.W.); 3Department of Computer Science, Mathematics Sul Ross State University, Alpine, TX 79830, USA; xiongnaixue@gmail.com; 4Center for AI Research, University of Agder, 4879 Grimstad, Norway; thanos.vasilakos@uia.no

**Keywords:** joint extraction, talking-head attention, Chinese medical texts, RoBERTa, health monitoring

## Abstract

Natural language processing (NLP) technology has played a pivotal role in health monitoring as an important artificial intelligence method. As a key technology in NLP, relation triplet extraction is closely related to the performance of health monitoring. In this paper, a novel model is proposed for joint extraction of entities and relations, combining conditional layer normalization with the talking-head attention mechanism to strengthen the interaction between entity recognition and relation extraction. In addition, the proposed model utilizes position information to enhance the extraction accuracy of overlapping triplets. Experiments on the Baidu2019 and CHIP2020 datasets demonstrate that the proposed model can effectively extract overlapping triplets, which leads to significant performance improvements compared with baselines.

## 1. Introduction

The emerging field of health monitoring aims to detect and prevent disease early by combining electronic medical records with health indicators; in general, the idea of health monitoring is to accurately extract key information from electronic medical records or medical texts [1]. The key information is usually represented in the form of triplets; therefore, as a branch of NLP [2], entity relation joint extraction is exploited to extract relational triplets between entities from texts. In the field of health monitoring, entity relation extraction helps to discover hidden knowledge and associations using massive health databases in order to optimize and improve health monitoring services [3].

The modeling of extracted entities and relations in early works primarily used the pipeline model [4]. The pipeline model separates the extraction task into two different subtasks, which then use two separate models to reduce computational costs. Although the pipeline model is flexible and simplifies the processing flow, there are many limitations on the management of the two independent models. First, a mistaken result during entity extraction has an impact on the relation extraction task, which can degrade the pipeline model’s performance. Second, the pipeline model ignores internal connections and dependencies between the entity recognition and relation extraction tasks [5]. Third, the pipeline model cannot provide an proper solution to the problem of overlapping entities in the named entity recognition task; this results in redundant information from candidate entities without relationships, which is detrimental to information extraction performance and imposes a considerable burden on optimal allocation of computing resources during the relationship extraction task. Therefore, joint models that extract entities and relations simultaneously have been proposed to overcome the above disadvantages.

In recent years, research on joint models has been predominantly based on English public datasets, and there have been few studies based on Chinese medical datasets. Therefore, joint extraction of entity relationship encounters many challenges with respect to Chinese medical texts. A primary challenge is that, because many medical entities involve multiple words, it is difficult for joint models to identify these entities. More importantly, the language of medical descriptions is quite complex, especially in electronic medical records of patients’ past and current medical history [6]. Furthermore, disease entities are associated with symptomatic entities. For example, the joint model extracts the triplet “cancer–symptom–pain”. The relationship placing “symptom” between “cancer” and “pain” can be matched correctly; hence, a more efficient method for joint extraction of entities and relations can be designed by utilizing position information and correlations between entities to improve the accuracy of the joint model.

In this paper, we propose a novel pointer network model based on joint entity and relation extraction for Chinese medical texts. Because of the excellent performance of RoBERTa on sentence encoding tasks, RoBERTa is first utilized as the encoding layer to encode sentences, thereby enhancing the connection between entities and extracting positional information in sentences. Next, based on the information encoded by RoBERTa, the position information is applied to strengthen the feature relationships in sentences and to fuse the different features occurring between entities. Finally, the talking-head attention mechanism is introduced to enhance interaction between relationships. Compared with baseline models on the Baidu2019 and CHIP2020 datasets, the proposed model is able to consider the semantic features in sentences while achieving improved accuracy.

In summary, the contributions of our work can be summarized as follows:We propose a joint entity and relation extraction model to effectively remedy the problem of overlapping triplets in Chinese datasets.The proposed model combines position information with a talking-head attention layer, adding additional semantic information and enhancing interaction between relationships.Our proposed model outperforms existing models in terms of the F1 value, realizing and improvement of 0.05 and 0.16 on the Baidu2019 and CHIP2020 datasets, respectively, and is able to extract entity relations in highly overlapping and complex sample datasets.

The rest of the paper is organized as follows. First, Section 2 introduces related works. In Section 3, the proposed joint model is described in detail. In Section 4, the datasets and the baseline models used for comparisons are described and the predictive performance of the proposed model is evaluated. Finally, we present our conclusions and discuss future research directions in Section 5.

## 2. Related Work

The existing relation extraction models in the field of medicine can be divided into pipeline models and joint models [7]. In pipeline models, the relation extraction task is divided into two independent subtasks and a model is constructed and trained for each subtask. On the other hand, joint models share parameters between the extraction results to simultaneously extract entities and relationships. In the field of medicine, early works addressed information extraction in a pipelined manner to enhance the accuracy of entity extraction and relation extraction. More specifically, the pipeline model has been applied to extract relational triplets in two separate steps: named entity recognition (NER) and relational classification (RC) [8].

In NER, during early sequence tasks the traditional methods are based on rules [9], dictionaries [10], and machine learning [11,12,13]. However, with the deepening of research on deep learning, these methods are becoming widely used in sequence modeling in the data-driven era, with examples including Long-Short-Term Memory (LSTM) [14], Condition-Random Field (CRF), and Bidirectional Encoder Representations from Transformers (BERT) [15]. With the development of deep learning, Liu et al. [16] applied the Bi-LSTM-CRF model to the Chinese clinical medical entity recognition system, utilizing CRF to achieve high micro-average F1-scores on multiple English datasets. LSTM can be used to recognize entities in specific formats; however, the accuracy of LSTM models on Chinese datasets is not acceptable. Therefore, Gridach et al. [17] introduced a novel neural network architecture that investigated a combination of LSTM, CRF, word embeddings, and character-level representation for recognition of biomedical named entities. Considering the characteristics of Chinese words segmentation, the model named Lattice LSTM [18] combined character information and lexical information for the first time while avoiding the influence of segmentation errors on LSTM. Zhao et al. [19] proposed a novel Chinese clinical named entity recognition method combining lattice LSTM and adversarial training; this method can improve the robustness of neural models by increasing perturbations in order to avoid the influence of segmentation errors. Li et al. [20] proposed a BERT–BiLSTM–CRF model for the medical field that simultaneously considers the characteristics of Chinese medical word segmentation and medical dictionary characters. The construction of the BERT–BiLSTM–CRF model has been applied to the field of EMR as well. Specifically, Gao et al. [21] introduced a BERT Chinese pre-training model able to automate feature selection, then combined BiLSTM and CRF to optimize the Chinese NER algorithm and applied the model to process an electronic medical record dataset. Because the LSTM model ignores context information, Kong et al. [22] sought to exploit the context information of short-term and long-term memory for the NER task by designing a simple attention mechanism that can improve training efficiency based on multiple Convolutional Neural Networks (CNN) in parallel. However, Multi-CNN models have difficulty capturing the spatial information between words. Therefore, Wang et al. [23] proposed an adversarial training LSTM–CNN system, which they called ASTRAL, to exploit position information between adjacent words. Unlike existing NER methods, ASTRAL improves the model structure and training process by introducing a Gated CNN to fuse the information of adjacent words.

In terms of medical RC, existing methods rely on medical features that can easily cause errors to accumulate during relation extraction, which then degrades the accuracy of feature extraction systems such as Recurrent Neural Networks (RNN) or other neural network-based methods. Hence, Fei et al. [24] proposed a BiLSTM–RNN model to learn the semantic features for relation extraction, and verified that LSTM–RNN can achieve better performance than LSTM on feature extraction. To improve the performance of RNNs, Zhang et al. [25] proposed a model that can automatically learn the features of sentence sequences by combining the RNN and CNN approaches for extracting biomedical relationships; however, this model fails to exploit the dependency types among words. To address this issue, a dependency-driven approach was proposed in [26] for relation classification using an attentive graph convolutional network (A-GCN), which applies a graph convolutional network-based attention mechanism to distinguish the importance of different dependencies between words. However, A-GCN lacks reliability when coding long sentences. Wang at el. [27] designed a structural block method to encode blocks associated with entities; the structural block method is able to eliminate noise caused by irrelevant parts of sentences to enhance the representation of relevant words. While the structural block method has the advantage of being independent of long sentence context, it only encodes the sequential tokens within a block boundary. Fortunately, BERT is able to encode long sentences, which has a significant impact on the natural language processing. To this end, Zhang et al. [28] introduced a clinical language model that used BERT for context pretraining, focusing on the importance of embedding in sentences. However, their clinical language model ignored the critical role of important phrase information. Therefore, Xu et al. [29] introduced a relational classification model based on BERT and a gated multi-window attention network (BERT–GMAN) to construct a key phrase classification network to obtain multi-granularity phrase information and exploit element-wise max-pooling to select the features of key phrases; BERT–GMAN showed greatly improved accuracy on the relational classification task. In addition, referencing subsequent improvements to BERT, the BioBERT model [30] has obtained excellent performance as a language representation model on multiple datasets. Nevertheless, the above-mentioned models ignore the connections between entity extraction and the relation extraction, which may lead to error propagation and decrease the accuracy of the extraction results.

Recently, many researchers have attempted to alleviate the problem of error propagation when jointly extracting entities and relations by exploiting complex semantic features with a single model. Based on complex semantic features, the initial joint model introduced the dependency syntax tree [31], which was able to effectively capture the features of sentences by stacking bidirectional tree-structured LSTM–RNNs on bidirectional sequential LSTM–RNNs. In this approach, the entities and relations are jointly represented to share parameters, allowing for entity and relation extraction in a single model. Based on previous work on dependency syntax trees [31], Katiyar et al. [32] replaced the dependency syntax tree with an LSTM network to determine different relations by comparing the similarity of entities with other entities. However, this approach ignores the entity boundary information in sentences. Hence, to enhance the accuracy of the pointer network in the process of decoding, Gu et al. [33] and Zeng et al. [34] introduced the copy mechanism to generate the relation and proposed the CopyNet and End2End Neural models, respectively. Among these, the CopyNet model uses a novel method of word generation based on the copying mechanism to choose proper sequences in the input and place them in the proper position in the output. On the other hand, the End2End Neural model utilizes the copy mechanism to jointly extract relational facts from sentences in the overlapping triplets class using different decoder strategies. Giannis et al. [35] proposed a multi-head selection model. This model considers the entity boundary information in sentences by introducing a CRF layer into the entity recognition task, thereby transforming the relationship extraction task into a multi-head selection problem. Although the multi-head selection model has demonstrated its effectiveness on multiple datasets, the model ignores robust generalizations during entity training as well as gaps between entities during the entity prediction process. To alleviate the model’s problems with robust generalizations and gaps, adversarial training mechanisms [36] and soft label embedding [37] have been proposed. Unlike the existing joint models, ETL-span [38] is a novel framework that can adequately capture semantic dependencies between different steps to remove noise between pairs of entities. Dixit et al. [39] directly extracted span-level features on the basis of the ETL–SPAN model; this model is able to pay attention to overlapping entities, avoiding erroneous information transmission cascade; unfortunately, the performance of this model in extracting triplets from long sentences is not satisfactory. To address the issue of relation extraction in long sentences, Eberts et al. [40] used BERT to encode long sentences and fuse multiple features before classifying relationships. In recent years, with the help of BERT, researchers have paid more attention to the connections between entities and relationships. Luo et al. [41] proposed a neural network framework called the ATT–BiLSTM–CRF model, which uses an attention mechanism for biomedical joint extraction. The ATT–BiLSTM–CRF model effectively strengthens the connection between the biomedical entity and the relationship. Hong et al. [42] proposed a joint entity relation extraction model based on a graph convolutional network (GCN) to effectively distinguish the interaction between entities and relationships. Lai et al. [43] designed a new multi-attentional mechanism to improve the performance of graph attentional networks, although this did not result in any improvement on overlapping relationships. Considering the issue of overlapping relationships, Wei et al. [44] proposed the Casrel model for relationship extraction, then merged the original sequence information using on BERT. To address the challenge of overlapping relationships, in this paper we propose a novel model that utilizes RoBERTa to encode sentences in the encoding layer and exploit the position information in order to enhance the feature representation of Chinese sentences. In addition, conditional layer normalization is combined with talking-head attention to alleviate the problem of overlapping triplets.

## 3. Proposed Joint Model

In this section, we present a description of the proposed joint extraction model motivated by the previous work of Wei et al. [44], in which the head entity is defined randomly for relation extraction without completely traversing all entities during entity extraction. The proposed model can identify all possible triplets in a sentence, even where a few triplets may share the same entities or the same relations. More specifically, the proposed model extracts triplets using two modules: a RoBERTa encoding module, and a cascade decoding module. In the RoBERTa encoding module, RoBERTa is applied to fully extract sentence features and identify connections between words in sentences. In the cascade decoding module, subject extraction is applied to find all possible subjects in the sentence. Then, relation–object extraction is applied to find all relevant relations and their corresponding objects. The cascade decoding module is designed with multi-level training objectives, simplifying the entity extraction process. The specific block diagram of the proposed model is shown in Figure 1. Among them, $a_{i}\left(i=1,2,3\right)$, $b_{j}\left(j=1,2,3,4\right)$, and $c_{k}\left(k=1,2,3\right)$ represent “pancreatic cancer”, “ultrasonic examination”, and “pancreatic masses”, respectively, in Chinese characters. The RoBERTa encoding module and cascade decoding module are elaborated upon in the following subsections.

### 3.1. RoBERTa Encoder

The RoBERTa encoder utilizes RoBERTa as an encoder to extract feature information from sentences. RoBERTa is a bidirectional coding representation algorithm based on the transformer algorithm for feature extraction and sentence modeling [45]. RoBERTa is able to learn deep representations by jointly conditioning on context, and can fine-tune additional output layers to perform efficiently on many downstream tasks.

In the encoder module, RoBERTa is applied to encode the sentence, which is then fed into subsequent decoder modules. Specifically, given a sentence x of length n, every word $x_{t}$ in the sentence can be transformed into a token embedding, segment embedding, and position embedding. Hence, the output $x_{t}$ of RoBERTa is as follows: (1)$$\begin{array}{c} x=\left\{x_{1},x_{2},\cdots ,x_{t},\cdots ,x_{n}\right\} \end{array}$$(2)$$\begin{array}{c} x_{t}=E_{S}+E_{T}+E_{P} \end{array}$$
where $E_{T}$ represents a word vector $E_{token}$, $E_{S}$ represents a segment vector $E_{segment}$, and $E_{P}$ represents a position vector $E_{position}$.

The detailed flow of RoBERTa in a Chinese sentence is shown in Figure 2, where $a_{i}\left(i=1,2,3\right)$ and $b_{j}\left(j=1,2,3,4\right)$ represent “pancreatic cancer” and “ultrasonic examination”, respectively, in Chinese characters. The twelve-layer transformer encoder of RoBERTa is utilized to encode the Chinese sentence with bidirectional coding representation, then three feature vectors are applied to reconstruct the sentence from the noisy data [46]. The feature vectors are then passed to the cascade module for subject extraction.

### 3.2. Cascade Decoder

The cascade module is adapted to extract triplets provided by the previous feature vectors. Specifically, the cascade decoder is divided into two cascaded steps, namely, subject extraction and relationship–object extraction. First, subject extraction detects subjects for each input sentence, including both head and tail entities. Second, relation–object extraction checks all possible relations to determine whether relations can be matched to the head and tail entities in the sentence. In addition, talking-head attention is utilized to obtain the relationships behind the conditional layer, which can enhance the accuracy of the cascaded steps. In the following subsections, the subject extraction and relationship–object extraction procedures are described in detail, as is the training algorithm used in the proposed model.

#### 3.2.1. Subject Extraction

In the subject extraction process, all possible subjects are recognized by directly decoding the feature vector h produced by the RoBERTa encoder. More specifically, this is essentially a binary classification problem that assigns each token a binary tag (0/1) that represents the start or end position of the identified subject by initializing a pointer network. Then, the subject is entered as a head entity in the module at the next level. The detailed operations used for subject extraction are as follows:(3)$$\left[\begin{array}{c} s^{start}\\ s^{end} \end{array}\right]=\sigma \left(\left[\begin{array}{c} W_{s}^{start}\\ W_{s}^{end} \end{array}\right]\times h+\left[\begin{array}{c} b_{s}^{start}\\ b_{s}^{end} \end{array}\right]\right)$$
where $s^{start}$ represents the probabilities of the start position of all subjects for the input sentence and $s^{end}$ represent the probabilities of the end position of all subjects for the input sentence. If the probability of a subject s exceeds a certain threshold, the corresponding tag is assigned a value of 1; otherwise, it has a value of 0. Here, h is the encoded representation for the input sentence, $W_{s}^{start}$ and $W_{s}^{end}$ represent the weight matrix of the start and end positions in the full connection layer, and are updated automatically, $b_{s}^{start}$ and $b_{s}^{end}$ are the respective offset vectors of the start and end positions, and σ is the sigmoid activation function used to map the output.

Next, the span of subject s in the input sentence x is optimized using the following likelihood function $p_{\theta }\left(s|x\right)$:(4)$$\begin{array}{c} p_{\theta }\left(s|x\right)=\prod_{t\in (start,end)}\prod_{i=1}^{L}{\left(s_{i}^{t}\right)}^{R_{1}^{t}}{\left(1-s_{i}^{t}\right)}^{R_{2}^{t}} \end{array}$$
where L is the length of the input sentence, $R_{1}^{start}$ and $R_{2}^{start}$ are marked as 1 and 0, respectively, if the subject start position is marked as 1 in the output start position sequence, $R_{1}^{end}$ and $R_{2}^{end}$ are marked as 0 and 1, respectively, if the subject end position is marked as 1 in the output end position sequence, and the parameter θ is $\left\{W_{s}^{start},b_{s}^{start},W_{s}^{end},b_{s}^{end}\right\}$. Moreover, the function $p_{\theta }\left(s|x\right)$ is exploited to evaluate the subject extraction performance.

For subject detection, the match principle for the nearest start–end distance is adopted to decide the span of subjects. For example, as shown in Figure 1, the matrices with the marked start and end positions of the three entities “pancreatic cancer”, “ultrasonic examination”, and “pancreatic mass” are obtained after the RoBERTa encoding layer. As the start token matches the nearest end token, the result of the first entity is “pancreatic cancer”. Based on the match strategy, the proposed model only considers those end tokens with positions behind the existing start token. More importantly, the match strategy can maintain the integrity of entities to the greatest extent possible.

#### 3.2.2. Relation–Object Extraction

Relation–object extraction simultaneously recognizes objects and their involved relations based on the previously obtained subjects. As shown in Figure 1, relation–object extraction consists of conditional layer normalization (CLN) and talking-head attention (THA). First, CLN is used to determine a specific category and randomly generate contexts based on this category. In particular, CLN [47] utilizes a fixed-length vector as a conditional scenario to incorporate the conditions **fi** and **fl** into normalization. Moreover, in CLN the feature h and two conditions are fused to combine relation features with the input entity features. Hence, the output $\hat{h}$ of CLN is as follows:(5)$$\begin{array}{c} \hat{h}=\left[\left(h-\mathrm{avg}\right)÷\mathrm{std}\right]\times \gamma +\beta \end{array}$$
where avg is the mean value of h, std is the standard deviation of h, and β and γ are two dynamic matrices that are influenced by the input subject in the sentence. Two different matrices that can be transformed by the different entities for initializing β and γ in the same dimension are exploited. In addition, two matrices β,γ and the feature h are combined to obtain the feature $\hat{h}$ that is affected by subject *s*. When merging these vectors, it is crucial that the dimension output by CLN remains consistent with the original pretraining model.

In order to exploit the parameters of entity recognition while improving the accuracy of relation extraction, the output matrix is spliced into CLN using the position information from RoBERTa, then the subsequent THA utilizes the matrix. The proposed model combines a feature $\hat{h}$ with the position information to obtain the matrix H, as follows:(6)$$H=\hat{h}+E_{P}$$

To enhance the effectiveness of feature extraction, the proposed model uses attention mechanisms. Compared with talking-head attention, multi-head attention [48] only focuses on the performance of each head, ignoring the relevance of the heads. The formulas for multi-head attention are as follows:(7)$$\begin{array}{c} Attention\left(Q,K,V\right)=softmax\left(\frac{QK^{T}}{\sqrt{d_{k}}}\right)V \end{array}$$
(8)$$\begin{array}{c} {\mathrm{head}}_{i}=Attention\left(QW_{i}^{Q},KW_{i}^{K},VW_{i}^{V}\right) \end{array}$$
(9)$$\begin{array}{c} \begin{array}{c} MultiHead\left(Q,K,V\right)=Concat\left({\mathrm{head}}_{1},{\mathrm{head}}_{2}\cdots ,{\mathrm{head}}_{h}\right)W^{O} \end{array} \end{array}$$
where Q, K, and V are converted from H, while $W_{i}^{Q}$, $W_{i}^{K}$, and $W_{i}^{V}$ represent the respective weight parameters of Q, K, and V for the *i*th calculation; moreover, $d_{k}$ represents the dimension of V, softmax is the softmax activation function, the single-head attention ${\mathrm{head}}_{i}$ is calculated using Equation (Equation 8), and $Attention\left(\cdot \right)$ in Equation (Equation 8) is illustrated in Equation (Equation 7). Finally, by repeating Equation (Equation 8) *h* times, the multiple attention $MultiHead\left(\cdot \right)$ is obtained based on the corresponding results for *h* times, where $W^{O}$ is the weight parameter and is automatically updated.

By linking the heads together, talking-head attention [49] can exploit more information from different representation subspaces at different positions. In addition, the information includes location information, syntax information, and other information. Hence, talking-head attention utilizes two additional learned matrices $\lambda_{i}^{W}$ and $\lambda_{i}^{L}$ to fuse head attention into talking-head attention. Therefore, the formulas for talking-head attention are as follows:(10)$$\begin{array}{c} A\left(Q,K\right)=\frac{QK^{T}}{\sqrt{d_{k}}} \end{array}$$
(11)$$\begin{array}{c} J_{i}=\lambda_{i}^{L}softmax\left(\lambda_{i}^{W}A\left(Q_{i},K_{i}\right)\right) \end{array}$$
(12)$$\begin{array}{c} \begin{array}{c} O=TalkingHead\left(Q,K,V\right)=Concat\left({J_{1}V}_{1}{{,\cdots ,J}_{h}V}_{h}\right) \end{array} \end{array}$$
where Q, K, and V are converted from H, $A(Q_{i},K_{i})$ represents the *i*th calculation of single-head attention, softmax is the softmax activation function, and $J_{i}$ indicates that single-head attention is associated with other attentions, where $\lambda_{i}^{W}$ and $\lambda_{i}^{L}$ can move information across attention heads by transforming the attention-logits and attention-weights, respectively. The output of talking-head attention TalkingHead(·) concatenates the calculation of attention for all heads.

Although the subjects are obtained by decoding the feature vector h during subject extraction, the head entity features are exploited during relation extraction to enhance the connection between the relations and the entities. Therefore, based on the features of the head entities in the sentences, the output of the relation extraction is as follows:(13)$$\begin{array}{c} \left[\begin{array}{c} r^{start}\\ r^{end} \end{array}\right]=\sigma \left(\left[\begin{array}{c} W_{r}^{start}\\ W_{r}^{end} \end{array}\right]*\left(O+v_{i}\right)+\left[\begin{array}{c} b_{r}^{start}\\ b_{r}^{end} \end{array}\right]\right) \end{array}$$
where $r^{start}$ and $r^{end}$ represent the probabilities of the respective start and end positions of all relations in the input sentence; the corresponding token are marked as 1 if the probability of the start and end positions exceeds a certain threshold, and are 0 otherwise. Moreover, $v_{i}$ represents the encoded representation vector between the start and end tokens of the *i*th subject detected in the subject extraction module, $W_{r}^{start}$ and $W_{r}^{end}$ represent the weight matrix of start and end positions relative to the relations, $b_{r}^{start}$ and $b_{r}^{end}$ are the respective offset vectors of the start and end positions of the relations, and σ is the sigmoid activation function.

More specifically, in order to achieve the fusion of O and $v_{i}$ in Equation (Equation 13), it is necessary to ensure that the dimension of the two vectors remains consistent during the relation–object extraction process. For each subject $v_{i}$, all subjects are traversed to extract triplets, while the subject is randomly selected for each sentence through the Casrel model [44]. Although the proposed model incurs higher computational cost during relation extraction, this results in improved accuracy during triplet extraction.

Next, the relation representations of the object *o* and subject *s* in the input sentence x are optimized using the following likelihood function $p_{\theta }\left(o|s,x\right)$:(14)$$p_{\theta }\left(o|s,x\right)=\prod_{t\in \left\{start,end\right\}}\prod_{i=1}^{L}{\left(r_{i}^{t}\right)}^{R_{1}^{t}}{\left(1-r_{i}^{t}\right)}^{R_{2}^{t}}$$
where L is the length of the sentence, $R_{1}^{start}$ and $R_{2}^{start}$ are marked as 1 and 0, respectively, if the object start position is marked as 1 in the output start position sequence, $R_{1}^{end}$ and $R_{2}^{end}$ are marked as 0 and 1, respectively, if the object end position is marked as 1 in the output end position sequence, and the parameter θ is $\left\{W_{r}^{start},b_{r}^{start},W_{r}^{start},b_{r}^{end}\right\}$. In addition, the function $p_{\theta }\left(o|s,x\right)$ is exploited to evaluate the relationship extraction performance.

As shown in Figure 1, for the output of each sentence a matrix is constructed to calculate the matching result between entities and relations. For example, the subject “pancreatic cancer” compares the relationships between “imageological examination”, “age”, and “clinical manifestation” with different objects, and all relations are traversed to determine the object that can be used to construct a triple with the subject “pancreatic cancer”. Finally, the two triplets “pancreatic cancer–imageological examination–ultrasonic examination” and “pancreatic cancer–clinical manifestation–pancreatic mass” are found.

For all training sets, the likelihood functions of entities and relations are optimized for each sentence x. The optimizer utilizes the Adam [50] loss function to maximize K by optimizing $p_{\theta }\left(s|x_{i}\right)$ and $p_{\theta }\left(o|{s,x}_{i}\right)$, which dynamically reduces the learning rate based on the number of times while increasing the model’s efficiency and effectiveness. The indicator K is written as follows:(15)$$\begin{array}{c} K=\sum_{i=1}^{\left|D\right|}\left[\sum_{s\in T_{i}}\log p_{\theta }\left(s|x_{i}\right)+\sum_{r\in T_{r}}\log p_{\theta }\left(o|{s,x}_{i}\right)\right] \end{array}$$
where |D| represents the cardinality of the training set, $T_{i}$ represents all subjects in the sentence, $T_{r}$ represents all relationships corresponding to the head entity, $p_{\theta }\left(s|x_{i}\right)$ is defined in Equation (Equation 4), and $p_{\theta }\left(o|{s,x}_{i}\right)$ is defined in Equation (Equation 14).

In the above, *K* is the key indicator that determines when the training process of the proposed model terminates. Specifically, if *K* is continuously updated until it reaches a steady state, then the training process is terminated. The training algorithm used for the proposed model is provided in Algorithm 1.
**Algorithm****1:** Training Algorithm of the Proposed Model**Input**: Training dataset *D*, training epochs *N***Output**: Optimal parameter *K***Pre-trained**: Use RoBERTa to obtain the encoded feature vector h **Initialize**: Transform Chinese sentences into vector representations and initialize the model parameters

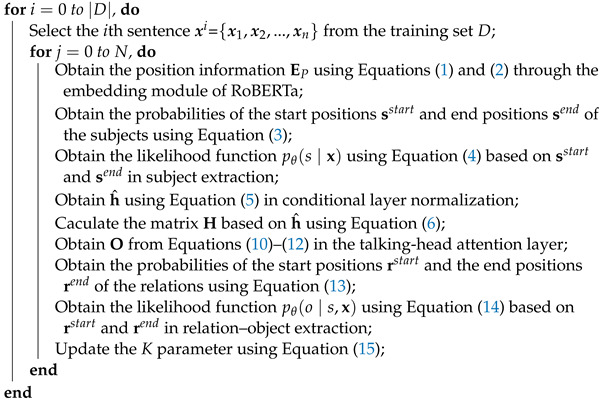
Return *K*;

## 4. Performance Analysis

In this section, our experiments are mainly introduced from three aspects. First, the data sources and components of the Baidu2019 and CHIP2020 datasets are explained. Second, the experimental setup is described in detail, including the implementation details, evaluation metrics, and experimental environment settings. Third, the superiority of the proposed model is verified by comparison with baseline models.

### 4.1. Datasets

The experiments were carried out on the Baidu2019 datasets and CHIP2020 datasets; the detailed description of the structures of these two datasets is as follows:Baidu2019 [51] contains sentences extracted from Baidu Baike and Baidu News Feeds; it is the largest Chinese information extraction dataset on the basis of schema, including more than 190,000 real-world Chinese sentences, more than 400,000 triplets, and 50 types of prespecified relations. To improve dataset availability, the dataset is divided into a training set and testing set using a certain proportion.CHIP2020 [52] is a Chinese medical dataset collected by the National Language Processing Laboratory of Zhengzhou University and the Key Laboratory of Computational Linguistics of the Ministry of Education of Peking University. It contains more than 17,000 Chinese medical sentences, more than 50,000 triplets, and 43 types of prespecified relations. The CHIP2020 dataset consists of diseases, symptoms, imaging tests, and other medical information. Moreover, the dataset is divided into a training set and testing set to enhance dataset standardization by establishing an official method.

The statistics of the two datasets are listed in Table 1 to further illustrate the characteristics of the Baidu2019 and CHIP2020 datasets. In addition, the datasets each include a training set and testing set that can be divided into three categories, as shown in Table 2, that is, Normal, Entity Pair Overlap (EPO), and Single Entity Overlap (SEO) [4,38]. Below, the results of different categories are analyzed in detail.

### 4.2. Experimental Setup

First, a number of the parameters in the proposed model are introduced. Specifically, the batch size was set to 8, the learning rate was set to $1\times 10^{-5}$, and the maximum length of the input sentence was set to 128. In addition, the number of heads for talking-head attention was 48, the number of transformer blocks was 12, and the size of the hidden state was 768. A stopping mechanism that ends the training process was adopted in the experiments. To measure the accuracy of the experimental results, the precision (pre), recall (rec), and F1-score are considered as the scoring functions, and can be written as shown below: (16)$$\begin{array}{c} pre=\frac{\mathrm{TP}}{\mathrm{TP}+\mathrm{FP}}\times 100\% \end{array}$$(17)$$\begin{array}{c} rec=\frac{\mathrm{TP}}{\mathrm{TP}+\mathrm{FN}}\times 100\% \end{array}$$(18)$$\begin{array}{c} F1=\frac{2\times pre\times rec}{pre+rec}\times 100\% \end{array}$$
where TP represents the number of correctly predicted triplets, FP represents the number of incorrectly predicted triplets, and FN represents the unpredicted triplets; pre explains the ratio of correctly predicted triplets to all predicted triplets, rec explains the ratio of correctly predicted triplets to all triplets in the datasets, and F1 provides a comprehensive evaluation of the results of precision and recall.

Next, the experimental environment is described concretely. As shown in Table 3, all models were implemented in TensorFlow-gpu 2.2.0 and trained on an Ubuntu 18.04 system with 32 GB of memory and an Nvidia 2080 GPU.

### 4.3. Results and Discussion

In this section, the results of our comparative experiments and ablation experiments are analyzed and discussed. Overlapping experiments were constructed on different types of sentences and the performance of the proposed model was compared with baselines. In addition, the results of parametric experiments on inference methods were applied to verify the effectiveness of the talking-head attention mechanism. Finally, the experiments selected different sentences in order to judge the accuracy of the proposed model in a case study.

#### 4.3.1. Comparative Experiment with Existing Research Works

In the experiments, several baseline methods were considered for comparisons:MultiR [55]: a multi-instance learning algorithm combining a sentence-level extraction model with a simple corpus-level module, which alleviates the problem of noise caused by labeling.CoType [56]: an extraction model that jointly utilizes text features and type labels when carrying out entity and relationship extraction, which considers the problem of overlapping.Multi-head selection [35]: a neural model that identifies multiple relations for each entity to perform relation extraction; it can simultaneously train an entity recognition module and relationship extraction module.Casrel [44]: a joint model designed as a novel cascade binary tagging framework derived from a principled problem formulation.ETL-span [38]: a specific label scheme that decomposes entity recognition and relationship extraction into several labeling problems to extract multiple triplets.

In order to more comprehensively verify the effectiveness of the proposed model, experiments were conducted on the Baidu2019 and CHIP2020 datasets by comparing it with baseline models. Table 4 shows a comparison of precision, recall, and F1 between the proposed model and the baseline methods on the Baidu2019 and the CHIP2020 datasets.

Table 4 shows the results on the Baidu2019 and CHIP2020 datasets. For the Baidu2019 dataset, it can be observed from Table 4 that the proposed model outperforms the best baseline models by 1.9% in triplet extraction. This improvement can be explained by the employment of the cascade decoder, which can accurately capture multiple relations. In addition, the proposed model achieves a 5.1% improvement in F1-score over the Casrel model on the Baidu2019 dataset. Unlike the Casrel model, the proposed model exploits RoBERTa to capture the semantic features in sentences and utilizes a talking-head attention mechanism to obtain more effective attention. The results show that the proposed model performs well on the task of feature extraction on Chinese datasets. Considering the results on the CHIP2020 dataset, it can be observed from Table 4 that the proposed model overwhelmingly outperforms all the baselines in terms of all evaluation metrics; in particular, it achieves a 14.6% improvement in F1-score over the ETL-Span model on the CHIP2020 dataset. Moreover, the pre-trained RoBERTa and talking-head attention are utilized to effectively extract single triplets and overlapping triplets from the Chinese medical dataset.

In addition, the results on these datasets show that there is a significant gap between the general field and the medical field when extracting triplets, as the proposed model has more difficulty dealing with overlapping triplets in the field of medicine. More precisely, as shown in Table 2, the Baidu2019 dataset mainly consists of the Normal and SEO classes, while the CHIP2020 dataset mainly includes the Normal and EPO classes. This inconsistent distribution of categories between the two datasets leads to better performance on the Baidu2019 dataset. Nonetheless, the proposed model achieves a smaller gap between the Baidu2019 and CHIP2020 datasets than the baseline models, which demonstrates its superior effectiveness on the task of extracting overlapping triplets in medical contexts.

#### 4.3.2. Ablation Experiments

Using the Baidu2019 and CHIP2020 datasets, our ablation experiments focused on the contribution of the RoBERTa encoder, CLN layer, and THA layer. Each time, a module in the RoBERTa encoder, CLN layer, or THA layer was removed to obtain the effect of that module on the proposed model. First, it can be seen from Table 5 that if the RoBERTa encoder is removed on the Baidu2019 and CHIP2020 datasets, the F1 score is reduced by 3.9% and 3%, respectively. These results verify that the RoBERTa encoder can effectively extract Chinese sentence features. Removing the CLN layer has have an apparent degradation effect on the F1-score as well, which indicates that combining the position information with the encoder feature is beneficial to the process of extracting triplets. When comparing with the models with and without the THA layer, it is clear that the THA layer provides a remarkable improvement in the F1-score, which demonstrates that talking-head attention can effectively improve the accuracy of overlapping triplet extraction.

#### 4.3.3. Analysis of Overlapping Triplets

To verify the ability of the proposed model to alleviate the problem of overlapping triplets, experiments were conducted on three categories of sentences and its performance was compared with the baseline models.

The results of the comparison between the proposed model and the baseline models on three categories of sentences are shown in Figure 3a–c. The results show that the proposed model achieves the best results on the Normal class, EPO class, and SEO class. Compared with the Casrel model, the F1-value improves by 10.9% and 11.9% on the EPO class of Baidu2019 and CHIP2020, respectively. Moreover, the F1 value improves by 6% and 16.2% for the SEO class. Indeed, among three categories of overlapping classes, the EPO and SEO classes are relatively complex collections of triplets. In contrast, the proposed model achieves consistently outstanding performance, especially for the EPO class, which shows that talking-head attention can alleviate problems on the EPO class by enhancing the relevance of features.

#### 4.3.4. Inference Method

To test the talking-head attention mechanism [49], experiments were structured comparing different heads to verify the reliability of the mechanism. Figure 4a,b shows the results of different heads on the Baidu2019 and CHIP2020 datasets.

It can be seen that talking-head attention efficaciously adjusts the trade-off between precision and recall with different choices of heads. It can be seen that when increasing the number of head from 1 to 48, the F1-score significantly increases by 2.4% and 2.8% on the Baidu2019 and CHIP2020 datasets, respectively. Furthermore, the proposed model works more effectively on both the Baidu2019 and CHIP2020 datasets as the number of heads increases. Due to the limitations of the experimental environment, 48 was chosen as the maximum number of heads.

#### 4.3.5. Case Study

To specifically and intuitively observe the ability of the proposed model in overlapping extraction, triplets were extracted from complex sentences selected from the CHIP2020 dataset and its performance was analyzed. For ease of understanding, the English annotations of the Chinese sentences selected from CHIP2020 dataset are shown in Table 6.

The first sentence of the selected Chinese sentences is shown in Table 6, which indicates all triplets of the entity ‘WD’. This sentence is classified as the SEO class, as the triplets of the sentence contain repeated entities. It can be be seen from Figure 5a that most of triplets were extracted accurately. More specifically, the proposed model correctly identified one disease and four drugs, and only missed on one drug. Moreover, it can be seen that the relationship between the entity ‘trientine’ and the entity ‘WD’ was not extracted because of the specific position of the entity ‘trientine’ in the sentence. In summary, the proposed model has more accurate extraction effects on the SEO class. On the second sentence shown in Table 6, classified as being from the EPO class, the results are shown in Figure 5b. This sentence contains more triplets, and has more than ten entities. Specifically, the proposed model correctly identified six diseases, three symptoms, and one examination, although it had one symptom is wrong. Furthermore, the wrong symptom (“fatty infiltration”) from the proposed model was extracted because the relationship between the symptom “fatty infiltration” and the disease “acute fatty liver during pregnancy” was misidentified. In the end, the results of the case study fully prove the excellent performance of the proposed model on Chinese medical sentences.

### 4.4. Engineering Applications

Intelligent medical treatment is a prominent future development trend in internet-based medical treatment. With the rapid development of information technology, more and more intelligent medical systems have been constructed to assist hospitals in diagnosing patients and even predicting diseases in advance. A health monitoring system is regarded as a kind of intelligent medical application scenario in the medical field, and is an important development for realizing medical data sharing and fusion. Due to the diversity and complexity of medical texts, existing research on the structuring of electronic medical records has mostly paid attention to exploiting deep learning models for completion intelligent medical tasks, such as medical entity recognition, medical relationship extraction, medical entity linking, medical entity alignment, etc. The structured electronic medical records are then applied for health monitoring and disease prediction. Medical entity recognition and medical relationship extraction are essential parts of the medical text structuring task. However, extracting the wrong triplets can have an enormous negative impact on the accuracy and universality of subsequent applications. In this regard, a more accurate deep learning model is proposed in this paper to complete the task of entity relation extraction. As shown in Figure 6, a data-driven approach is employed to structure electronic medical record data. More specifically, electronic medical records from hospitals and health indicators from devices are exploited to predict patients’ condition.

As mentioned above, in order to further improve the accuracy of health monitoring, our next work will focus on training more types of data, including technical medical terms, unstructured crawler data, etc. In addition, as the deep learning model has visible shortcomings, such as the lack of robustness, the lack of annotated data, the few-shot learning method could be incorporated into the proposed model to reduce the dependence of the model on labeled data. In addition, this method would improve the efficiency of extraction process while saving labor cost. The proposed model could have great significance for the construction of intelligent medical system.

## 5. Conclusions and Future Work

This article highlights the advantages of artificial intelligence technology for health monitoring. Specifically, we propose a novel model for joint extraction of entities and relationships to improve the accuracy of health monitoring. The proposed model transforms joint extraction into a binary tagging problem. We introduce RoBERTa to fully extract sentence features. Furthermore, we exploit conditional layer normalization in the decoder to combine entities with relationships. Talking-head attention is applied to strengthen the interaction between entity recognition and relation extraction. Thus, the proposed model can simultaneously extract different triplets from sentences and alleviate the problem of overlapping triplets. We conducted complex experiments on two Chinese datasets to demonstrate the effectiveness of the proposed model. Our experimental results on the Baidu2019 and CHIP2020 datasets show that the proposed model outperforms baseline models. Ablation experiments were used to demonstrate the importance of each module. In summary, the experiments show that the proposed model can effectively extract overlapping triplets and has better performance than existing methods.

In the future, different technologies can be further explored to extract information efficiently in Chinese sentences. First, the accuracy of the proposed model on the SEO class needs to be improved in order to increase the accuracy of the healthcare monitoring system. In order to solve the problem of low efficiency on SEO class identification, one option is to investigate different attention fusion methods. A second issue is that this model does not effectively identify medical texts, as special medical sentences are complex. Hence, for special medical sentences, medical dictionaries could be combined with medical sentences to enhance the model’s performance on extracting overlapping triplets in medical contexts.

## Figures and Tables

**Figure 1 sensors-23-04812-f001:**
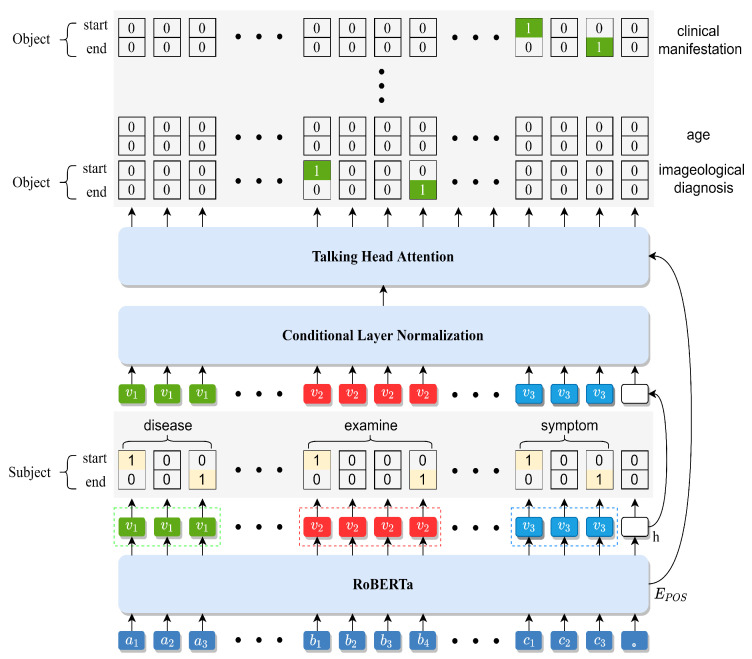
Block diagram of the proposed model.

**Figure 2 sensors-23-04812-f002:**
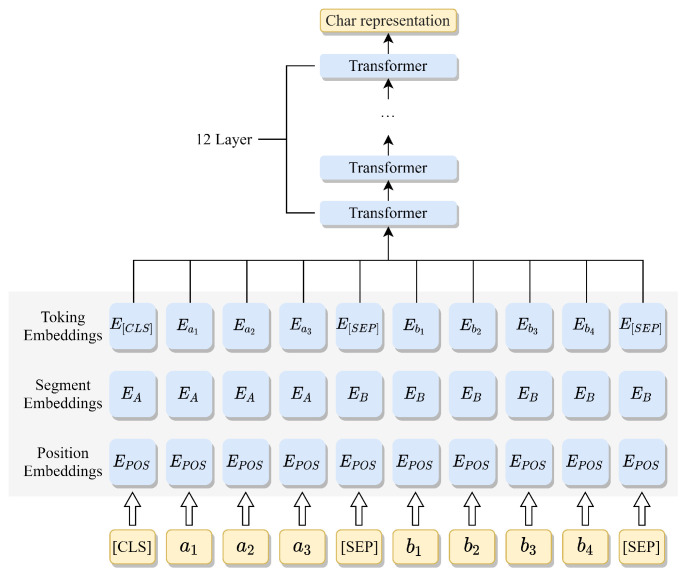
Block diagram of RoBERTa applied to a Chinese sentence.

**Figure 3 sensors-23-04812-f003:**
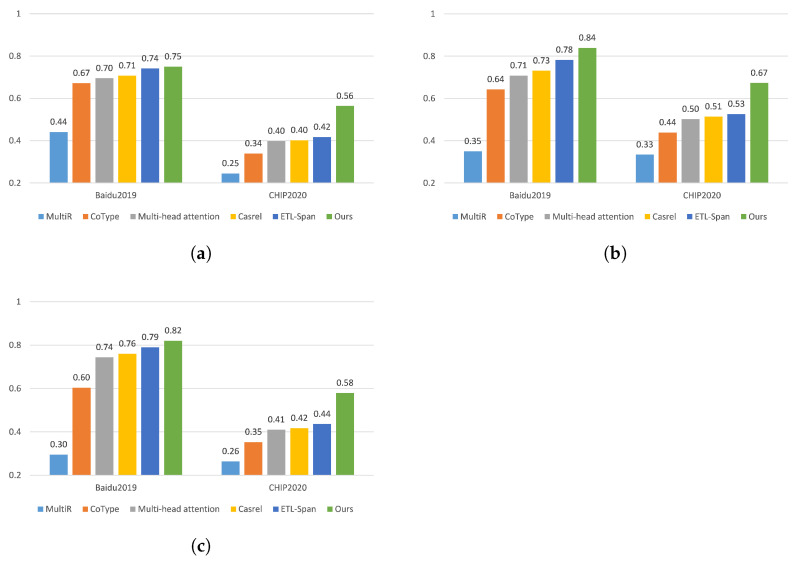
F1-score when extracting relational triplets from sentences on the different classes: (**a**) Normal class; (**b**) EPO class; (**c**) SEO class.

**Figure 4 sensors-23-04812-f004:**
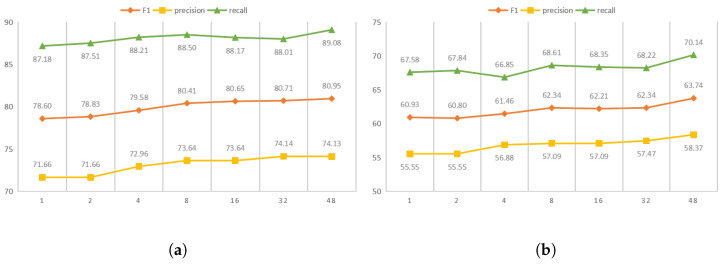
The performance on different datasets with different heads: (**a**) Baidu2019 dataset and (**b**) CHIP2020 dataset.

**Figure 5 sensors-23-04812-f005:**
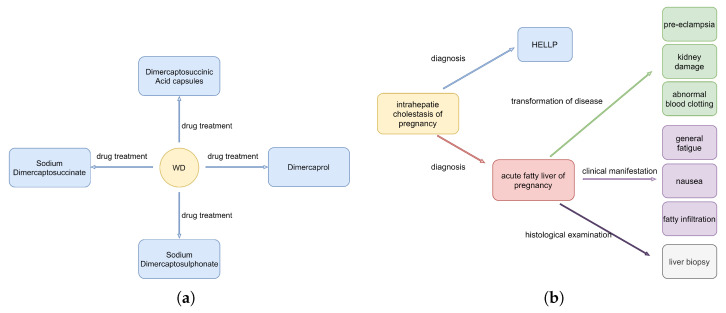
The triplets extracted by the proposed model in the case study: (**a**) case_1 and (**b**) case_2.

**Figure 6 sensors-23-04812-f006:**
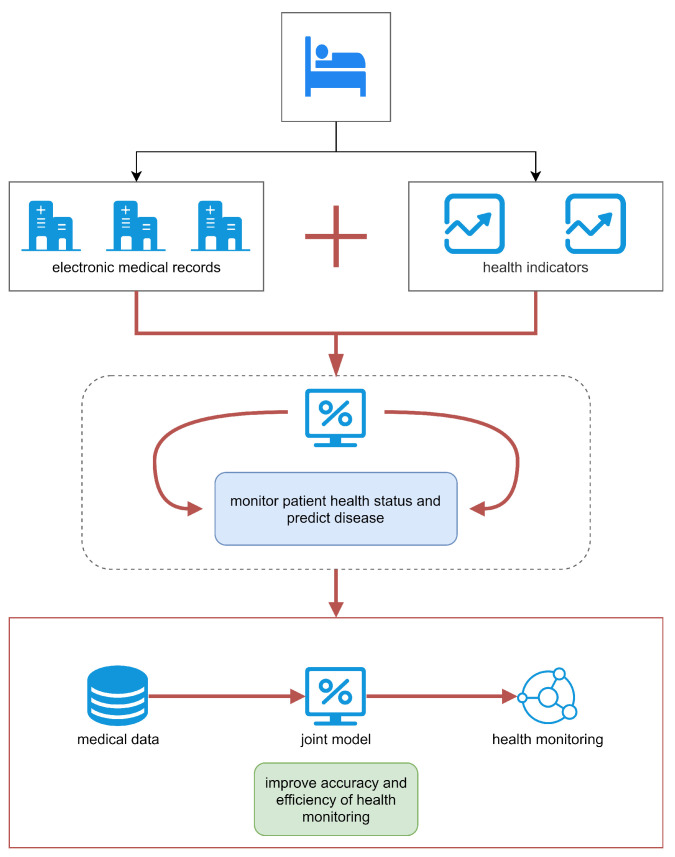
An application diagram of the proposed model in health monitoring.

**Table 1 sensors-23-04812-t001:** Statistics of the two datasets.

Statistics	Baidu2019		CHIP2020	
	Train	Test	Train	Test
sentence	172,983	21,626	14,339	3585
riplets	363,895	45,558	43,660	10,626
relations		50		43

**Table 2 sensors-23-04812-t002:** Statistics of different categories of triplets in the datasets.

Category	Baidu2019		CHIP2020	
	Train	Test	Train	Test
Normal	80,310	9984	5724	1496
EPO	19,049	2385	6937	1655
SEO	73,596	9257	1678	434

**Table 3 sensors-23-04812-t003:** Experimental environment settings.

Item	Environment
Operating system	Ubuntu 18.04.5 LTS
CPU	i7-8700 @3.20 GHz
GPU	NVIDIA GeForce RTX 2080Ti
Memory	31 G
Python version	3.7
TensorFlow [53] version	TensorFlow-gpu 2.2.0
Transformers [54] version	3.1.0

**Table 4 sensors-23-04812-t004:** Comparisons with different methods on the Baidu2019 and the CHIP2020 datasets.

Method	Baidu2019			CHIP2020		
	Precision	Recall	F1	Precision	Recall	F1
MultiR [55]	0.634	0.389	0.482	0.261	0.378	0.312
CoType [56]	0.729	0.703	0.716	0.344	0.497	0.41
Multi-head attention [35]	0.764	0.712	0.737	0.412	0.572	0.471
Casrel [44]	0.800	0.720	0.758	0.42	0.581	0.48
ETL-Span [38]	0.779	0.801	0.790	0.41	0.633	0.494
Ours	0.801	0.838	0.809	0.566	0.767	0.64

**Table 5 sensors-23-04812-t005:** Results of the ablation experiments.

Method	Baidu2019	CHIP2020
	F1	F1
Ours	0.809	0.64
-RoBERT	0.77	0.61
-CLN	0.778	0.62
-THA	0.782	0.61

**Table 6 sensors-23-04812-t006:** Results of the case study on different sentences.

Sentence	Text	Our Model
case_1	Trientine is also a complexing agent, which can promote the excretion of copper. It is sometimes used as a first-line drug in WD patients with neurological symptoms. It is effective in all types of patients, and the general dose is 40–50 mg/(kgd). Other copper drugs: Dimercaprol (because of side effects have been less), Sodium Dimercaptosuccinate, Dimercaptosuccinic Acid capsules and Sodium Dimercaptosulphonate and other heavy metal chelate agents.	see Figure 5a
case_2	Intrahepatie cholestasis of pregnancy(HELLP) is acute fatty liver of pregnancy. The patient developed the classic symptoms of general fatigue, nausea, pre-eclampsia, abnormal blood clotting and kidney damage. Liver biopsy showed fatty infiltration, but biopsy is rarely performed during diagnosis.	see Figure 5b

## Data Availability

Not applicable.

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
