# Peer review of "A Joint Extraction System Based on Conditional Layer Normalization for Health Monitoring"

_sensors, 2023, doi:10.3390/s23104812_

Round 1

Reviewer 1 Report

I would like to thank the authors for this contribution. The study presents an interesting application of NLP in the medical domain. The methodology is largely clear, and the results are robustly validated. However, please consider the comments below in the next version.

(1)

In order to provide a better understanding of the current state of NLP in healthcare, I recommend positioning the introduction appropriately within the context of recent studies that have discussed the role of NLP in the healthcare context. For example:

https://doi.org/10.5220/0010414508250832

(2)

Please cite the original reference of BERT:

Devlin, J., Chang, M., Lee, K., & Toutanova, K. (2019). BERT: Pre-training of Deep Bidirectional Transformers for Language Understanding. In Proceedings of the Annual Conference of the North American Chapter of the Association for Computational Linguistics (NAACL-HLT).

(3)

If the Hugging Face repository was used, please ensure to cite their reference.

Wolf, T., Debut, L., Sanh, V., Chaumond, J., Delangue, C., Moi, A., ... & Rush, A. M. (2019). Huggingface's transformers: State-of-the-art natural language processing. arXiv preprint arXiv:1910.03771.

(4)

Likewise, please cite the references of libraries used such as TensorFlow.

(5)

Please cite the references of the datasets used.

(6)

Please provide further elaboration on the possible limitations of the study results to give readers a more complete understanding of the study's findings.

Overall, I appreciate the authors' efforts and look forward to seeing an improved version of this study.

The quality of language is generally good, though it could be improved at some paragraphs that seem relatively wordy.

Reviewer 2 Report

In this paper, the author propose a model for joint extraction of entities and relations. This method  combines the conditional layer normalization with the talking-head attention mechanism to strengthen the interaction between entity recognition and relation extraction. At the same time , Using location information to improve the accuracy of overlapping triplet extraction. In my opinion, although the results seem good, they still need to be modified. Some specific comments can be seen below.

1.       There are problems with references. For example, The individual references are too old, have no reference value, and the format of references and articles is inconsistent.

2.       There are many problems with the layout of the chart. For example, Table 4 is too large for the content of the article, and the layout in Figure 3 is inconsistent with the content of the article and is too low in clarity. These need to be completely revised.

3.         For this article, a detailed parameter description is necessary. Although some parameters are given in this article, they are not detailed enough and should be described more specifically so that the reader can understand them better.

4.       In the experimental part, the benefits of the proposed model performance can be better demonstrated by adopting comprehensive model evaluation indicators and graphs, such as ROC and curve charts. It is recommended to add department evaluation indicators and corresponding dynamic graphs.

Overall, I believe that the presentation and content of this thesis needs to be revised.

In this paper, the author propose a model for joint extraction of entities and relations. This method  combines the conditional layer normalization with the talking-head attention mechanism to strengthen the interaction between entity recognition and relation extraction. At the same time , Using location information to improve the accuracy of overlapping triplet extraction. In my opinion, although the results seem good, they still need to be modified. Some specific comments can be seen below.

1.       There are problems with references. For example, The individual references are too old, have no reference value, and the format of references and articles is inconsistent.

2.       There are many problems with the layout of the chart. For example, Table 4 is too large for the content of the article, and the layout in Figure 3 is inconsistent with the content of the article and is too low in clarity. These need to be completely revised.

3.         For this article, a detailed parameter description is necessary. Although some parameters are given in this article, they are not detailed enough and should be described more specifically so that the reader can understand them better.

4.       In the experimental part, the benefits of the proposed model performance can be better demonstrated by adopting comprehensive model evaluation indicators and graphs, such as ROC and curve charts. It is recommended to add department evaluation indicators and corresponding dynamic graphs.

Overall, I believe that the presentation and content of this thesis needs to be revised.

Round 2

Reviewer 2 Report

The paper  can  be accepted now.

The paper  can  be accepted now.